# Immune Checkpoint Inhibitors and Mismatch Repair Status in Advanced Endometrial Cancer: Elective Affinities

**DOI:** 10.3390/jcm11133912

**Published:** 2022-07-05

**Authors:** Alessandro Rizzo

**Affiliations:** Struttura Semplice Dipartimentale di Oncologia Medica per la Presa in Carico Globale del Paziente Oncologico “Don Tonino Bello”, I.R.C.C.S. Istituto Tumori “Giovanni Paolo II”, Viale Orazio Flacco 65, 70124 Bari, Italy; rizzo.alessandro179@gmail.com

**Keywords:** immunotherapy, endometrial cancer, immune checkpoint inhibitors, mismatch repair deficiency, dostarlimab, pembrolizumab, lenvatinib

## Abstract

Since endometrial cancers (ECs) are frequently TMB-H and MSI-H/dMMR tumors, this element has provided the rationale for testing immune checkpoint inhibitors (ICIs), which have recently emerged as a potential game-changer. However, several questions remain to be addressed, including the identification of patients who may benefit from the addition of ICIs as well as those who do not need immunotherapy. In the current paper, we provide an overview of the clinical development of immunotherapy in advanced or recurrent EC, discussing the role of MMR and the “elective affinities” between ICIs and this predictive biomarker in this setting.

## 1. Introduction

Endometrial cancer (EC) represents an important cause of cancer-related death worldwide, showing a 25% increase over the past decade and representing the most common gynecological cancer in the developed world, as well as the only gynecological malignancy with rising mortality [1]. Unfortunately, women with advanced or recurrent disease have limited treatment options, and to tackle this unmet medical need, several hormonal treatments, cytotoxic chemotherapies, and targeted therapies have been investigated over the last decade [2,3]. Hormone therapy is the preferred initial treatment in case of low-grade (grade 1 or 2) tumors, not rapidly growing, while six cycles of carboplatin and paclitaxel given every three weeks are the current standard of care in advanced/recurrent disease, being associated with a median progression-free survival (PFS) of approximately one year. Immune checkpoint inhibitors (ICIs) have been tested in advanced or recurrent EC, with the emergence of immunotherapy recently representing a breakthrough in previously treated patients [4]. However, an important proportion of ECs receiving ICIs do not benefit from immunotherapy due to several reasons, including DNA mismatch repair (MMR) status, with the identification of biomarkers and elements able to predict as well as to impair response to ICIs, which remains a fundamental issue in this setting [5,6].

The advent of genome sequencing has provided an unprecedented amount of information in the landscape of EC [7]. In fact, several genetic alterations have been identified, with *PIK3CA* and *TP53* mutations that have been frequently observed in both serous and endometrioid malignancies [8]. In particular, *PIK3CA* mutations have been highlighted in approximately 50% of endometrioid tumors and 40% of serous ECs, while *TP53* mutations in 75 and 12% of endometrioid and serous tumors, respectively. In addition, the Cancer Genome Atlas project has further defined the molecular scenario of EC, based on genomic, proteomic, and transcriptomic data, and has identified four different subgroups of malignancies: (1) polymerase (POLE)-mutant/ultra-mutated; (2) microsatellite instability-high (MSI-H); (3) copy number low (CNL); and (4) copy number high (CNH) [9,10]. This molecular classification has been shown to also have a predictive value, with some of these subgroups—MMR-deficient and POLE-mutated tumors—having been reported to predict a lack of benefit from adjuvant chemotherapy; in addition, the molecular characterization has a clear impact on the risk of stratification since, for example, all POLE-mutated tumors, defined by the pathogenic mutation in the exonuclease domain of polymerase epsilon, are considered low-risk malignancies, irrespective of stage and grade [11].

In the current paper, we provide an overview of the clinical development of immunotherapy in advanced or recurrent EC, discussing the role of MMR and the “elective affinities” between ICIs and this predictive biomarker in this setting. We performed research on PubMed/Medline, the Cochrane library and Scopus using the keywords “endometrial cancer” OR “endometrial carcinoma” AND “immune checkpoint inhibitors” OR “immunotherapy” OR “PD-1 inhibitor” OR “atezolizumab” OR “nivolumab” OR “durvalumab” OR “pembrolizumab” OR “dostarlimab” OR “mismatch repair” OR “mismatch repair proficient” OR “mismatch repair deficient”.

## 2. Endometrial Cancer and MMR

MMR deficiency (dMMR) has been reported in up to 30% of all ECs, something that supports the routine test of MMR in all patients [12]. In approximately 90% of cases, at least one MMR gene presents a somatic mutation, while in 10% of cases, germline mutations are detected [13]. Since the MMR pathway corrects DNA replication errors that lead to the incorporation of the wrong nucleotide as well as nucleotide insertions and/or deletions, dMMR results in the accumulation of mutations and may lead to an MSI-H phenotype [14]. Specifically, when at least one MMR protein is inactivated by epigenetic changes, a malignancy is defined as dMMR. In addition, germline mutations in MMR genes defines a condition called Lynch syndrome, a form of inherited dMMR accounting for approximately 2% of ECs [15].

EC represents a particularly attractive target for checkpoint inhibitor-based treatment for several reasons, and a strong biological rationale has supported the exploration of immunotherapy in EC patients [16]. It is well known that, in physiological conditions, checkpoint pathways prevent excessive T-cell activation that may result in loss of self-tolerance [17]; for example, Programmed Cell Death Ligand 1 (PD-L1) and Cytotoxic T-Lymphocyte Antigen 4 (CTLA-4) regulate the stimulation of the immune microenvironment. Malignancies are able to adapt to these responses and to exploit checkpoint pathways to promote tumor growth, and thus, ICIs can help to reactivate T-cell function, leading to killing of the tumor cells [18]. In addition, tumor features play a key role in impacting the responsiveness to ICIs. In fact, tumors with a high tumor mutational burden have been suggested to be more responsive to immunotherapy; these tumors are classically considered as immunologically “hot” and are infiltrated with cytotoxic cytokines and tumor-infiltrating T cells (Figure 1) [19].

Conversely, immunologically “cold” tumors are characterized by a lack of tumor-infiltrating lymphocytes and cytokines with cytotoxic activity. EC represents the solid tumor with the greatest percentage of MSI-H/dMMR cases, ranging from 25 to 30%, according to recent studies [20]. Since MSI-H/dMMR tumors carry from 10 to 100 times as many somatic mutations compared with MMR-proficient (pMMR)/microsatellite stable (MSS) tumors, it is readily apparent that this higher mutational burden may increase immune activity. In addition, dMMR malignancies present prominent lymphocyte infiltrates, priming them for immune-mediated activity. Based on these biological premises, the MSI-H/dMMR phenotype has emerged as a predictive biomarker for checkpoint inhibitor therapy in several malignancies, including EC [21].

## 3. Immune Checkpoint Inhibitors Monotherapy

Several ICIs have been tested as monotherapy for EC patients, including the PD-L1 inhibitor avelumab, the PD-1 inhibitor pembrolizumab, the PD-L1 inhibitor durvalumab, and the anti-PD-1 agent dostarlimab. Among these, pembrolizumab and dostarlimab have been approved by regulatory agencies for MSI-H/dMMR patients.

Following preliminary data from an early-phase trial and a pooled analysis, the anti-PD-1 agent pembrolizumab has been tested in the KEYNOTE-158 trial [22,23]. In this open-label, multicohort, basket trial, the investigators enrolled 79 previously treated patients with MSI-H EC; these patients received pembrolizumab 200 mg every three weeks, and the primary endpoint of this trial was overall response rate (ORR) by independent-central review, with the duration of response (DoR), PFS, overall survival (OS), and safety also assessed as secondary endpoints [22,23]. A clinically meaningful ORR of 48% was highlighted, with complete response and partial response detected in 11 (14%) and 27 (34%) cases, respectively. In addition, a long duration of response was reported, with an estimated DoR of 68% at three years. Regarding treatment-related adverse events, the safety profile of pembrolizumab was consistent with previous literature studies, with all grade toxicities and grade 3–4 events reported in 76 and 12% of EC patients, respectively [22,23]. Pruritus, fatigue, and diarrhea were the most frequently observed all-grade toxicities (24, 32, and 16%, respectively). In terms of immune-mediated adverse events, hypothyroidism (14%) and hyperthyroidism (8%) were the most common [22,23].

The PD-1 inhibitor dostarlimab was assessed in the GARNET trial [24,25]. This phase I, single-arm study investigated the role of dostarlimab (TSR-042) in several tumor types, including EC. The dMMR cohort (*n* = 103) included patients who had progressed on or after platinum doublet therapy, and had received two or less than two prior lines of treatment for recurrent or advanced disease. The ORR was 46% in dMMR EC, with complete response and partial response observed in 10.7 and 34% of cases, respectively; the disease control rate (DCR) was 59% [24,25]. In addition, the investigators explored the ORR according to the number of prior lines of treatment. Interestingly, the ORR was higher in patients with one prior line of therapy (50%, with complete response in 9.1% and partial response in 40.9% of cases) compared with ECs previously treated with at least two lines of treatment (ORR of 35.9%, complete response and partial response of 12.8 and 23.1%, respectively). The DCR was also higher in EC patients with one prior line (63.6 versus 46.2% for heavily pretreated subjects) [24,25]. Overall, dostarlimab was well tolerated, with grade 3 or more treatment-related adverse events observed in 13.5% of cases and all-grade toxicities in 9.5%. Diarrhea, fatigue, and nausea were the most frequently observed all-grade events (15.9, 13.5, and 12.7%, respectively). In terms of all-grade immune-mediated adverse events, hypothyroidism (5.6%) and diarrhea (4.8%) were the most common [24,25]. There were two (1%) reports of grade 3 or more treatment-related colitis, and no reports of grade 3 or more pneumonitis. No deaths were attributed to dostarlimab.

Unfortunately, results regarding the role of ICIs monotherapy in pMMR/MSS ECs have been different, and an overall limited activity has been observed. In fact, single-agent pembrolizumab has shown an ORR of 11% in 107 pMMR EC patients enrolled in the KEYNOTE-158 study [26,27]; similarly, the GARNET trial highlighted an ORR of 13% in pMMR EC. Activity has been even lower in two other clinical trials investigating durvalumab and avelumab in the same patient population, observing a disappointing ORR of 3 and 6%, respectively. At the same time, it is worth noting that a crucial point would be the identification of more reliable predictors of response to ICI monotherapy. For example, two exploratory analyses of KEYNOTE-158 and GARNET have provided some interesting data regarding the role of tumor mutational burden (TMB) as a predictive biomarker. Pembrolizumab reported high ORR in TMB-high (TMB-H) EC compared with non-TMB-H tumors, with 10 mutations/megabase determined as the threshold for TMB-H status. Similarly, TMB-H tumors reported higher response to dostarlimab in GARNET (44.9 and 13.0% in TMB-H and TMB-low patients, respectively) [26,27]. Interestingly, in TMB-H patients, the ORR was 44.8 and 45.5% in dMMR and pMMR ECs, respectively, something that may suggest a comparable activity in these two groups. Thus, these data provide interesting evidence regarding the role of MMR as a predictive biomarker in this setting, that would certainly deserve further investigation, along with the role of TMB.

Beyond pembrolizumab and dostarlimab, other ICI monotherapies have been tested for EC patients. Among these, a phase II trial assessed the activity of the PD-L1 inhibitor durvalumab in pMMR tumors progressing after one to three lines of chemotherapy and dMMR patients whose disease had progressed after zero to three lines of chemotherapy [28]. The ORR in the latter cohort was 47% (17/36), including six cases of complete responses and 11 partial responses, versus 3% in pMMR patients. The median PFS in dMMR and pMMR ECs was 8.3 months and 1.8 months, respectively. In another phase II trial, Konstantinopoulos and colleagues explored the role of avelumab in pMMR and dMMR EC [29]; the first cohort was closed at the first stage due to futility, while the dMMR group met the predefined primary endpoint of four objective responses after the accrual of 17 patients.

## 4. Immune-Based Combinations

With regard to combination therapies including ICIs, two areas are considered to be of particular interest: the combinations of immunotherapy plus antiangiogenic agents and those including ICIs with cytotoxic chemotherapy, since both combinations have the potential to play a synergistic action [30,31].

The most important data regarding immune-based combinations with antiangiogenic agents have been reported in the recently published KEYNOTE-775 phase III trial comparing the efficacy and safety of lenvatinib plus pembrolizumab versus the treatment of the physician’s choice (paclitaxel or doxorubicin) in previously treated EC patients with advanced, metastatic, or recurrent disease [32]. The primary endpoints of KEYNOTE-775 were PFS and OS, with ORR, health-related quality of life (HRQoL), and safety assessed as secondary outcome measures. The study met its primary endpoints: the combination of lenvatinib plus pembrolizumab was associated with a statistically significant and clinically meaningful improvement in terms of PFS in pMMR and all-comers. In particular, the median PFS was 6.6 months and 3.8 months in pMMR patients receiving lenvatinib–pembrolizumab and chemotherapy, respectively (HR, 0.60; 95% CI, 0.5–0.72; *p* < 0.0001) [32]. These results were also mirrored in the all-comers treated with the immune-based combination, with a median PFS of 7.2 months (versus 3.8 months for chemotherapy; HR, 0.56; 95% CI, 0.47–0.66; *p* < 0.0001). Similarly, the combination reported a statistically significant OS benefit for pMMR, with a median OS of 17.4 months and 12.0 months in EC patients receiving lenvatinib–pembrolizumab and chemotherapy, respectively (HR, 0.68; 95% CI, 0.56–0.84; *p* < 0.0001) [32]. In addition, lenvatinib plus pembrolizumab showed its superiority over doxorubicin or paclitaxel in terms of median OS in the all-comers (median OS of 18.3 months versus 11.4 months). An interesting exploratory analysis was conducted in the dMMR patient population: in these ECs, an even higher PFS and OS benefit was reported. In fact, the median PFS was 10.7 months and 3.7 months in dMMR patients receiving lenvatinib plus pembrolizumab and the treatment of the physician’s choice, respectively (HR, 0.36; 95% CI, 0.23–0.57; *p* < 0.0001). Similarly, the median OS was not reached in dMMR ECs treated with the immune-based combination, while it was 8.6 months in those receiving chemotherapy (HR, 0.37; 95% CI, 0.22–0.62; *p* < 0.0001). Of note, lenvatinib–pembrolizumab reported an ORR of 40% in dMMR patients, with complete response and partial response observed in 13.8 and 26.2%, respectively. In terms of safety, dose reductions due to treatment-related adverse events were particularly common in the immune-based combination arm (66.5 versus 12.9% in the chemotherapy group), and unfortunately, 33% of patients experienced treatment discontinuation due to toxicities [32]. The results of the ongoing phase III study, the ENGOT-en9/MK-7902-001/LEAP-001 trial, comparing pembrolizumab plus lenvatinib versus paclitaxel–carboplatin in newly diagnosed EC patients are highly awaited, and will help to understand whether the immune-based combination will be moved to the first-line setting. Enrollment started in April 2019 and the estimated primary completion date is April 2023. The primary endpoints are PFS and OS. A key point to consider is that, according to KEYNOTE-775, combination therapy not only improves the response rate in pMMR, but also in dMMR—something that poses further questions on the role of MMR as a predictor of response in this setting.

Another treatment strategy under development includes the combination of immunotherapy plus cytotoxic chemotherapy [33,34]. Among the ongoing clinical trials, the ATTEND trial is randomly assigning EC patients to carboplatin–paclitaxel or atezolizumab plus carboplatin–paclitaxel, followed by placebo or atezolizumab maintenance (Table 1). The primary outcome measures are PFS and OS. Likewise, the NRG-GY018 is comparing pembrolizumab plus carboplatin–paclitaxel versus carboplatin plus paclitaxel plus placebo in patients with treatment-naïve, advanced, or recurrent EC, and the RUBY phase III study is assessing dostarlimab plus chemotherapy.

## 5. Conclusions

As reported, ECs have a relatively high proportion of TMB-H and MSI-H/dMMR tumors, something that has provided the rationale for the development of treatment with ICIs. In recent years, the dMMR phenotype has emerged as a predictive biomarker for immunotherapy, and ICIs such as pembrolizumab and dostarlimab have shown clinically meaningful activity in dMMR EC patients as monotherapy. Conversely, the efficacy of single-agent ICIs appears limited for pMMR tumors, and further efforts are needed to identify the predictors of response in this setting. In addition, another important point to highlight is related to Lynch genes and POLE/POLD1 mutations, since these genetic aberrations have been suggested to be involved in the response to immunotherapy. Conversely, dMMR ECs may present DNA methylation or silencing of one of these genes, and frequently do not respond to ICIs. At the same time, since some pMMRs have also been reported to benefit from ICIs, additional predictive biomarkers should be considered for patient selection [35,36,37]. Thus, candidates for PD-1 blockade may extend beyond POLEm and dMMR ECs; additional factors such as tumor grade and combination of TIL levels and expression of checkpoint proteins may need to be considered for the optimization of treatment. The immune-based combination of lenvatinib plus pembrolizumab has provided potentially practice-changing results in previously treated patients, regardless of MMR status, and several ongoing ICIs trials have the potential to continue to provide evidence to further modify this treatment scenario, and to clarify the “elective affinities” between EC immunotherapy and MMR.

## Figures and Tables

**Figure 1 jcm-11-03912-f001:**
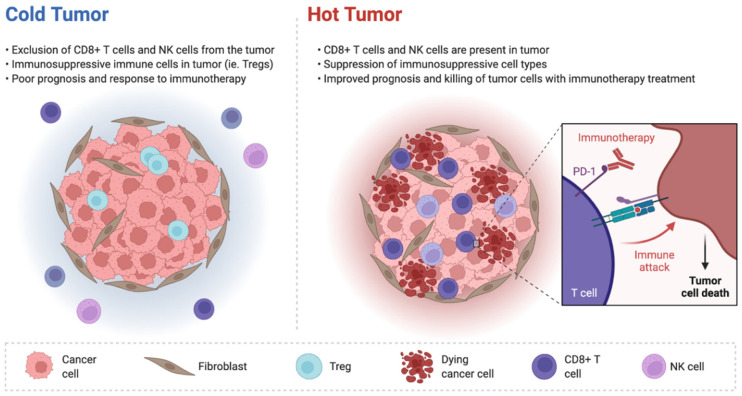
Schematic figure representing immunologically “cold” and “hot” tumors. Hot tumors are infiltrated by CD8+ T cells and NK cells, while immunologically cold tumors lack these cells.

**Table 1 jcm-11-03912-t001:** Some ongoing clinical trials assessing immunotherapy in endometrial cancer.

NCT ID	Phase	Treatment Arms
NCT03951415	II	Durvalumab plus olaparib
NCT02912572	II	Avelumab or avalumab plus axitinib
NCT03603184	III	Atezolizumab plus carboplatin plus paclitaxel versus placebo plus carboplatin plus paclitaxel
NCT03835819	II	Mirvetuximab Soravtansine (IMGN853) plus pembrolizumab

## Data Availability

Not applicable.

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
