# Peer review of "Immune Checkpoint Inhibitors and Mismatch Repair Status in Advanced Endometrial Cancer: Elective Affinities"

_jcm, 2022, doi:10.3390/jcm11133912_

Round 1

Reviewer 1 Report

The current review has addressed well the overview of the clinical development of immunotherapy in advanced endometrial cancer, discussing the role of mismatch repair status. The introduction is comprehensive and informative. Study design and presentation of the clinical trials are quite appropriate for the scope of the manuscript. Although there is a need for a review on this topic, there are some issues and questions that should be addressed in the study.

1-  For dMMR/MSI-H advanced endometrial cancer, a single-agent ICI could be a highly effective treatment However, some clinical trials indicates which cases are more likely to respond well: those containing mutations in genes known as Lynch genes and also some with mutations in POLE/POLD1 (“ultra-hypermutation”genes). In contrast, the majority of dMMR endometrial cancers have silencing or DNA methylation of one of these genes, MLH1, and do not appear to be as responsive to single-agent immune therapy. Endometrial cancers with MMRd or POLEm have been suggested as promising candidates for anti PD-1/PD-L1 therapy, whereas a recent study also reported that a subgroup of pMMR endometrial cancer may also benefit from this therapeutic approach, suggesting additional factors need to be considered for patient selection. For dMMR cases showing a loss of MLH1/PMS2 protein expression, MLH1 promoter methylation testing should then be performed, and an assessment of TMB and POLEm may be considered. Taken together, MLH1 promoter methylation and mutations in POLE/POLD1 should be discussed. (doi: 10.3389/fonc.2021.640018. eCollection 2021, https://doi.org/10.1002/cncr.34024 and doi: 10.1002/cncr.34025. )

2- Some recent and relevant studies could enrich your manuscript.

  • Clinical activity of durvalumab for patients with advanced mismatch repair-deficient and repair-proficient endometrial cancer. A nonrandomized phase 2 clinical trial (doi: 10.1136/jitc-2020-00225 )
  • Phase II Study of Avelumab in Patients With Mismatch Repair Deficient and Mismatch Repair Proficient Recurrent/Persistent Endometrial Cancer( 1200/JCO.19.01021)
  • Efficacy and safety of nivolumab in Japanese patients with uterine cervical cancer, uterine corpus cancer, or soft tissue sarcoma: Multicenter, open‐label phase 2 trial (doi: 1111/cas.14148)

3-  It would be valuable to tabulate the summary of ongoing studies on immune checkpoint inhibitor combination in advanced endometrial cancer in clinicaltrials.gov. Some examples include:

-DOMEC trial - NCT03951415- Durvalumab and Olaparib

-NCT02912572- Avelumab alone and in combination with Talazoparib or Axitinib

-atTEnd trial- NCT03603184 -carboplatin and paclitaxel plus atezolizumab vs placebo

-NCT03835819 - Mirvetuximab Soravtansine (IMGN853) and Pembrolizumab

4- P1 Line 42-43 Define four different subgroups clearly (1,2,3 and 4 or a,b,c, and d)

5- There are a number of grammatical/spelling errors which need correcting. Some examples include:

     P1 Line 15 -Please correct “In the current Review”

     P5 Line 203 -  Please correct “in dMMR EC patiens”

Author Response

Dear Reviewer, 

Thank you so much for the time spent revising our work.

We have extensively modified the manuscript, according to your suggestions.

  1. We further highlighted the topic of predictive biomarkers in this setting, as suggested, and the papers you recommended have been included (purple).
  2. We included the studies above, as suggested (blue).
  3. We modified accordingly, and we added a table (green).
  4. We modified accordingly.

Thank you again for your valuable comments.

We hope the revised paper will better suit the journal.

Reviewer 2 Report

The main purpose of the article is to illustrate that different MMR status has certain selectivity for the application of ICI in EC patients. However, it has already been recommended pembrolizumab for non-MSI-high/dMMR tumors in EC NCCN guildlines. The current review mainly lists the results of existing clinical trials. Therefore, fresh comments are suggested to enrich the article .

Author Response

Dear Reviewer,

Thank you for your comments and the time spent revising our paper.

We have modified several parts of the manuscript, as suggested, as you could find in the revised paper, and we added several comments and points.

Thank you again. We hope the revised manuscript will better suit the journal.

Reviewer 3 Report

This review covers immune checkpoint inhibitors and mismatch repair status in advanced endometrial cancer. Some comments should be addressed for improvement.

1. Introduction: detailed descriptions including specific survival rates according to the conventional therapies should be added.

2. Immune checkpoint inhibitors monotherapy and Immune-based combinations: Please add tables including cited studies and outcomes (e.g. ORR, OS etc.) according to the therapies for better readability. 

3. Please provide specified criteria for included studies such as search strategy if applicable.

Author Response

Dear Reviewer, 

Thank you so much for your valuable comments and the time spent revising our paper.

We extensively modified the manuscript and we considered your suggestions. Our changes have been reported in color.

We hope the revised manuscript will better suit the journal.

Round 2

Reviewer 1 Report

I am satisfied that the authors have addressed all of my previous concerns about the article. 

Author Response

Dear Reviewer,

Thank you again for the time spent revising our paper.

Reviewer 2 Report

l  Ambiguous expression or writing problem: Line 47-49 "four different subgroups" have confusing serial numbers "1), 2) and d)", please check it.

l  Some conclusions drawn by the author have been described in the article. For example, in the last paragraph of the subsection "2 Endometrial cancer and MMR", it is clear that "MSI-H/dMMR has emerged as a predictive biomarker for EC". The core idea of the authors ​​has been clarified in previous research

l  The results cited in Line 128-142 are that among patients with TMB-H, the overall response rates to single-agent immunotherapy in patients with dMMR and pMMR were similar (44.8% and 45.5%, respectively). It supports "the role of tumor mutational burden (TMB) as a predictive biomarker", but does this result want to say that MMR is not a reasonable biomarker, and TMB is more suitable.

l  Tumor mutational burden (TMB), tumor microsatellite instability (MSI), and tumor mismatch-repair deficiency (dMMR) are all characteristics that reflect the characteristics of tumor cells themselves. The 3 indicators are not completely equivalent. In this article, these 3 concepts are not very clear and are replaced as synonyms, which is inappropriate.

l  Paragraph 4, the combination therapy not only improves the response rate in pMMR but also in dMMR, this does not indicate whether MMR is suitable as a predictor of treatment.

Overall, the authors discuss the rationality of MMR as a new biomarker for EC, both single and combined. The efficacy of dMMR in single drug is significantly better than that of pMMR, and TMB can also be used as a predictor of single drug efficacy. The predictive effect of TMB has nothing to do with the status of MMR; the combination of drugs has little to do with MMR status, but the article does not mention it. What indicators can be used to predict the efficacy of the combination therapy? Evidence may be relatively weak (only 2 clinical trials described dMMR as status in EC patients)

Author Response

Dear Reviewer,

Thank you again for the time spent revising our paper.

We further modified the revised manuscript, as you could find below.

Our changes are reported in red.

Reviewer 3 Report

Most of comments were addressed. I hope that this paper contribute to understand Immune Checkpoint Inhibitors and Mismatch Repair Status in Advanced Endometrial Cancer.

Author Response

(The authors gave the same response as above.)
